# T cell stiffness is enhanced upon formation of immunological synapse

Philipp Jung[1†], Xiangda Zhou[2†], Sandra Iden[3], Markus Bischoff[1], Bin Qu[2,4*]

[1]Institute for Medical Microbiology and Hygiene, Saarland University, Homburg, Germany; [2]Department of Biophysics, Center for Integrative Physiology and Molecular Medicine (CIPMM), School of Medicine, Saarland University, Homburg, Germany; [3]Cell and Developmental Biology, School of Medicine, Center of Human and Molecular Biology (ZHMB), Saarland University, Homburg, Germany; [4]Leibniz Institute for New Materials, Saarbrücken, Germany

**Abstract** T cells are activated by target cells via an intimate contact, termed immunological synapse (IS). Cellular mechanical properties, especially stiffness, are essential to regulate cell functions. However, T cell stiffness at a subcellular level at the IS still remains largely elusive. In this work, we established an atomic force microscopy (AFM)-based elasticity mapping method on whole T cells to obtain an overview of the stiffness with a resolution of ~60 nm. Using primary human CD4[+] T cells, we show that when T cells form IS with stimulating antibody-coated surfaces, the lamellipodia are stiffer than the cell body. Upon IS formation, T cell stiffness is enhanced both at the lamellipodia and on the cell body. Chelation of intracellular Ca[2+] abolishes IS-induced stiffening at the lamellipodia but has no influence on cell-body-stiffening, suggesting different regulatory mechanisms of IS-induced stiffening at the lamellipodia and the cell body.

**\*For correspondence:**
bin.qu@uks.eu

[†]These authors contributed equally to this work

**Competing interest:** The authors declare that no competing interests exist.

## Introduction

T cells are activated by the engagement of T-cell receptors (TCRs) with the matching antigen on the target cells. Consequently, CD3 molecules, one key component of the TCR complex, transduce the signal to activate downstream pathways leading to formation of a tight contact between T cells and target cells termed the immunological synapse (IS) (*Bromley et al., 2001*). During IS formation, the adhesion molecule LFA-1 (lymphocyte function-associated antigen 1) binds its ligand ICAM1 (Intercellular Adhesion Molecule 1) on the target cells to seal and stabilize the IS (*Bromley et al., 2001*). Upon T cell activation, intracellular Ca[2+] concentration is drastically enhanced via Ca[2+] influx. Ca[2+] serves as an essential second messenger in T cells to regulate their activation, proliferation, and effector functions (*Trebak and Kinet, 2019*). Artificial IS can be also induced between T cells and functionalized surfaces (*Chin et al., 2020*; *de la Zerda et al., 2018*).

Recently, it has been revealed that mechanical properties play a significant role in modulation of T cell functions (*Chin et al., 2020*; *Harrison et al., 2019*; *Rossy et al., 2018*). Cytotoxic T cells optimize their killing function via applying mechanical forces (*Basu et al., 2016*), where the mechanic output is coordinated with release of cytotoxic granules (*Jin et al., 2019*; *Tamzalit et al., 2019*). Force generation of T cells requires a sustained elevation of intracellular calcium and integrity of a functional F-actin network (*Basu et al., 2016*; *Hui et al., 2015*; *Jin et al., 2019*). Notably, even when the actin-cytoskeleton is perturbed, application of periodical mechanical forces linked to TCRs can still induce Ca[2+] influx (*Basu et al., 2016*). Furthermore, responsiveness of T cells to stimuli is elevated on stiffer substrates (40–50 kPa) relative to their softer counterparts (< 12 kPa) (*Majedi et al., 2020*; *Zhang et al., 2020*). Recent findings indicate that LFA-1 engagement plays an essential role in regulating T

cell responsiveness to substrate stiffness (*Wahl et al., 2019*). However, how the stiffness of T cells per se is regulated upon IS formation is still not well understood.

Various methodologies have been developed to investigate cell stiffness, for example micropipette aspiration, microplate-based rheometry (*Desprat et al., 2006*; *Hochmuth, 2000*), high-throughput real-time deformation cytometry (*Fregin et al., 2019*), optical tweezers (*Feng et al., 2017*; *Killian et al., 2018*), and atomic force microscopy (AFM). The differences of these methods were discussed elsewhere (*Wu et al., 2018*). AFM uses a mechanical probe (also called cantilever) to measure stiffness at defined regions, which is also applicable on living cells. For this purpose, the cantilever is usually brought into oscillation in close proximity to the cell surface to allow periodical contacts between the cantilever and the surface. This setting enables the measurement of local interacting forces at each point of a defined scanning grid, allowing the examination of height profiles, and to measure mechanical properties with nanometer spatial and pN force resolution (*Thewes et al., 2015*). These features enable a precise determination of local stiffness at the site of interest, which is pivotal for characterization of mechanic properties at the IS (*de la Zerda et al., 2018*).

In this work, we established an AFM-based method to simultaneously determine the surface profile and stiffness of live T cells. Human CD4$^+$ T cells were immobilized on glass coverslips via adhesion molecule LFA-1 with or without CD3/CD28 activation. We found at the lamellipodia, the stiffness was significantly higher than that at the cell body. Remarkably, upon formation of IS induced by CD3/CD28 stimulation, T cells were substantially stiffened at the cell body as well as at the lamellipodia. By chelating intracellular Ca$^{2+}$ with EGTA-AM, we identified that Ca$^{2+}$ is involved in regulation of this IS formation-induced T cell stiffening at the lamellipodia but not at the cell body.

## Results

### T cells are stiffened upon IS formation

To examine the stiffness of T cells upon IS formation, we established a method to investigate living T cells on functionalized coverslips by AFM based Peak Force Quantitative Nanoscale Mechanical Characterization (Peak Force QNM) (*Berquand et al., 2010*). We first functionalized polyornithine-coated coverslips with anti-LFA-1, anti-CD3 and anti-CD28 antibodies. CD28 is a co-stimulatory molecule, essential for sustained T cell function (*Esensten et al., 2016*). Without anti-LFA-1 antibody T cells did not attach to the surface, which is a prerequisite to determine cell stiffness by AFM. We first settled Jurkat T cells, which present features of effector CD4$^+$ T cells (*Abraham and Weiss, 2004*), on the functionalized coverslip for 15 min. Subsequently, a quarter of each cell was investigated by Peak Force QNM at a resolution of ~60 nm between measurements to create a high-density map of local elastic moduli (Young's moduli determined by Derjaguin-Muller-Toporov fit). We observed that after making contact with antibody-coated coverslips, Jurkat T cells formed flattened spreading structures (hereinafter referred to as lamellipodia), which dynamically changed over time (*Figure 1A*). Similar dynamic changes were also observed with primary human CD4$^+$ T cells with the same experimental setting (*Figure 1B*). This morphology of lamellipodia detected by AFM resembles findings made by scanning electron microscopy and immunostaining (*Saitakis et al., 2017*; *Schoppmeyer et al., 2017*). In addition, complete height profiles were obtained, showing that T cells were flattened on the functionalized surface (*Figure 1C*), suggesting a functional artificial IS was formed as observed in previous studies (*Pattu et al., 2011*; *Qu et al., 2011*). To simplify, the artificial IS formed between T cells and functionalized surface is referred to as IS in this work.

Next, we compared the stiffness of primary human CD4$^+$ T cells settled either on anti-CD3/anti-CD28/anti-LFA-1 antibodies (hereinafter referred to as full antibody set)-coated surfaces or on control surfaces (anti-LFA-1 antibody-coated). The latter condition induces only T cell attachment to the surface. Our results show that on the full antibody set-decorated surfaces, primary human CD4$^+$ T cells exhibited a significantly enhanced stiffness at both the lamellipodial regions and the cell body relative to their counterparts on control surfaces (*Figure 2A and B*), with a fold change of 2,5-fold and twofold (*Figure 2C and D*), respectively. The stiffness of the lamellipodia of primary human CD4$^+$ T cells was on average also higher than that of the cell body (*Figure 2E and F*), although this difference was not statistically significant under full-antibody condition (*Figure 2E*). Taken together, our findings show that T cells are stiffened upon IS formation, where stiffness of lamellipodia is higher than the cell body.

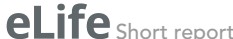

**Figure 1.** Representative time points of lamellipodial dynamics at the IS. (**A, B**) Dynamic changes of the lamellipodium of a Jurkat T cell (**A**) or a human primary CD4$^+$ T cell (**B**) during IS formation on a αLFA-1+αCD3+αCD28 antibody-coated coverslip. The height profile, examined by Peak Force QNM, is displayed (upper panel: 3D view, lower panel: top view). Exemplary dynamic parts are highlighted by arrowheads. (**C**) Height profile of a whole primary human CD4$^+$ T cell during IS formation on a αLFA-1+αCD3+αCD28 antibody-coated coverslip. One representative cell from at least three independent experiments is shown.

We next analyzed whether the local stiffness at the lamellipodium was correlated with positioning or topology of the microstructure. To this end, we selected individual spots on the lamellipodial regions, especially at tips/edges, close to the cell body, and in between (*Figure 2—figure supplement 1*). Here, we observed very similar Young's moduli for these regions from the same cell, with (αLFA-1+αCD3+αCD28) or without (αLFA-1) the IS (*Figure 2—figure supplement 1*). This finding

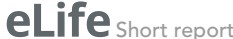

**Figure 2.** The stiffness of T cells increases upon activation. Height profiles and corresponding elasticity maps, Young's moduli, and the respective fold changes of human primary CD4+ T cells. Either glass coverslips (**A–F**) or PDMS (**G–K**) were applied as the functionalized surfaces. (**H–J, L**) Young's modulus of primary T cells on full antibody (αLFA-1+αCD3+αCD28)-functionalized glass (n = 11, the same dataset as in (**D**) full antibody), PDMS (2.5 MPa, n = 10), or PDMS (400 kPa, n = 5) substrates. The Mann-Whitney test (**C**), (**D**), (**H**), and (**I**), the Wilcoxon matched-pairs signed rank test (**E**), (**F**), and (**J**) or Mann-Whitney-U-test (**L**) was used for analyzing statistical significance. The results were presented as mean ± SEM, from 7 to 12 cells per condition as shown in the plots (LFA-1 vs full anybody-set) from six independent experiments for (**A–F**) (six donors), from 10 cells (four independent experiments/four donors) for PDMS condition (2.5 MPa) in (**G–J**), or from 5 cells (two independent experiments/two donors) for PDMS condition (400 kPa). Source data please refer to *Figure 2—source data 1*. For height profiles and elasticity maps of each value and condition, refer to *Figure 2—*

*Figure 2 continued on next page*

*Figure 2 continued*

*figure supplements 3–6*. For representative Force-Distance Curves, refer to *Figure 2—figure supplement 7*.

The online version of this article includes the following source data and figure supplement(s) for figure 2:

**Source data 1.** Original values of stiffness shown in *Figure 2*.

**Figure supplement 1.** Local stiffness at lamellipodia is not influenced by positioning or topology.

**Figure supplement 1—source data 1.** Stiffness of T cells measured on funcationalized surfaces.

**Figure supplement 2.** Stiffness of substrates.

**Figure supplement 2—source data 1.** Source data of stiffness of uncoated and coated substrates.

**Figure supplement 3.** Height profiles and elasticity maps (Young's modulus) of primary T-cells on αLAF-1-functionalized glass.

**Figure supplement 4.** Height profiles and elasticity maps (Young`s modulus) of primary T-cells on full antibody (αLFA-1+αCD3+αCD28)-functionalized glass.

**Figure supplement 5.** Height profiles and elasticity maps (Young`s modulus) of primary T-cells on full antibody (αLFA-1+αCD3+αCD28)-functionalized PDMS substrate with 2.5 MPa.

**Figure supplement 6.** Height profiles and elasticity maps (Young`s modulus) of primary T-cells on full antibody (αLFA-1+αCD3+αCD28)-functionalized PDMS substrate with 400 kPa.

**Figure supplement 7.** Exemplary Force-Distance Curves during Elasticity mapping of primary T cells on full antibody (αLFA-1+αCD3+αCD28)-functionalized glass.

indicates that the elasticity within the lamellipodia is rather a universal property, which is not affected by the topology or the position of the microstructures with an area of a few hundred $nm^2$. Notably, cytoskeleton serves as scaffolds at the protrusions such as actin filaments or microtubules. The diameters of these filamentous structures (~ 5–25 nm) is, however, far below the resolution (~ 60 nm) of our experimental setting. Thus, we cannot exclude the possibility that along these cytoskeletal filaments the elasticity may differ from the neighboring structures.

We further examined whether the stiffness of lamellipodia measured by this method could be influenced by stiffness of the substrate rigidity. In order to address this, we first used Polydimethylsiloxane (PDMS) substrates with a stiffness of 2.5 MPa. The stiffness of the lamellipodia of primary T cells was characterized on a full body-functionalized PDMS surface using the same settings as for glass coverslips (*Figure 2G*). No significant difference in stiffness was observed between glass and PDMS surface for the cell body (*Figure 2H*) or at the lamellipodium (*Figure 2I*). On the PDMS surface, stiffness of lamellipodia was significantly higher than that of cell body and the fold change was very similar to that on glass coverslips (compare *Figure 2J* with *Figure 2E*). Next, we used even softer PDMS substrates with an elasticity of ~400 kPa, on which the IS could be formed and lamellipodial regions could be analyzed (*Figure 2K*). Notably, no significant changes were identified in the Young's moduli of the lamellipodia of T cells formed on the full antibody-functionalized softer PDMS substrates (400 kPa; *Figure 2K*), compared to the ones measured on full antibody-functionalized glass or stiffer PDMS substrates (2.5 MPa) (*Figure 2L*). These findings indicate that the Young's moduli determined for the lamellipodial regions in the range of hundreds of kPa were not markedly influenced by the stiffness of the substrates used. In addition, to further reduce the risk of a potential influence of the backing material stiffness on our measurements, we applied a peak force threshold of 700 pN during elasticity mapping, which caused an indentation of 21 ± 2.2 nm into the cell body and 17 ± 4.7 nm into the lamellipodia, respectively. Since the lamellipodia displayed a mean height of 144 ± 69.7 nm we can largely exclude an impact of the substrates on the Young's moduli examined for the lamellipodia.

During our analyses, we noticed that the stiffness on cell-free functionalized glass coverslips was in a range of several hundred kPa to several MPa, which is much lower than the expected stiffness of glass (which lies in the range of GPa). To elucidate the reason for this, we first examined the stiffness of uncoated glass coverslips. Since the expected Young's moduli are in the range of GPa, we used a stiffer cantilever with a spring constant of 0.8 N/m and experimental settings suitable to characterize hard substrates (For further details please see Materials and methods). Young's moduli of uncoated glass coverslips were around 1 GPa (*Figure 2—figure supplement 2A*), verifying that with these experimental conditions our system is able to detect the stiffness in the GPa range. Next, we determined the Young's moduli of uncoated substrates with a soft cantilever and the experimental settings applied to characterize the stiffness of T cells. The average Young's moduli of uncoated glass coverslips

were around 18 MPa with the highest values around 20 MPa (*Figure 2—figure supplement 2B*), which is clearly below the stiffness values determined with the stiffer cantilever and suggestively the upper limit of the Young's moduli that can be determined by these experimental conditions optimized for determination of T cell stiffness. Nevertheless, the Young's moduli of uncoated PDMS substrates (2.5 MPa and 400 kPa) were in the expected range, indicating that up to 2.5 MPa the stiffness can be precisely determined by the corresponding experimental conditions. Next, we analyzed the substrate rigidity for each value of our existing datasets from *Figure 2*. We found that the functionalization of stiffer substrates (i.e. glass and 2.5 MPa PDMS) with αLFA-1 or αLFA-1+αCD3+αCD28 created surfaces with drastically reduced Young's moduli of around 600–700 kPa (*Figure 2—figure supplement 2B*), while the stiffness of the αLFA-1+αCD3+αCD28 functionalized softer PDMS (~400 kPa) was comparable to the uncoated PDMS (400 kPa) (*Figure 2—figure supplement 2B*). This indicates that antibody-functionalization substantially decreases the apparent stiffness of the cantilever-accessible upmost surface layer of stiff substrates.

## T cell stiffening is regulated by calcium

We next sought for the underlying mechanism regulating T cell stiffening triggered by TCR-activation. Since the cytoskeleton plays an important role in maintaining cell stiffness (*Gavara and Chadwick,*

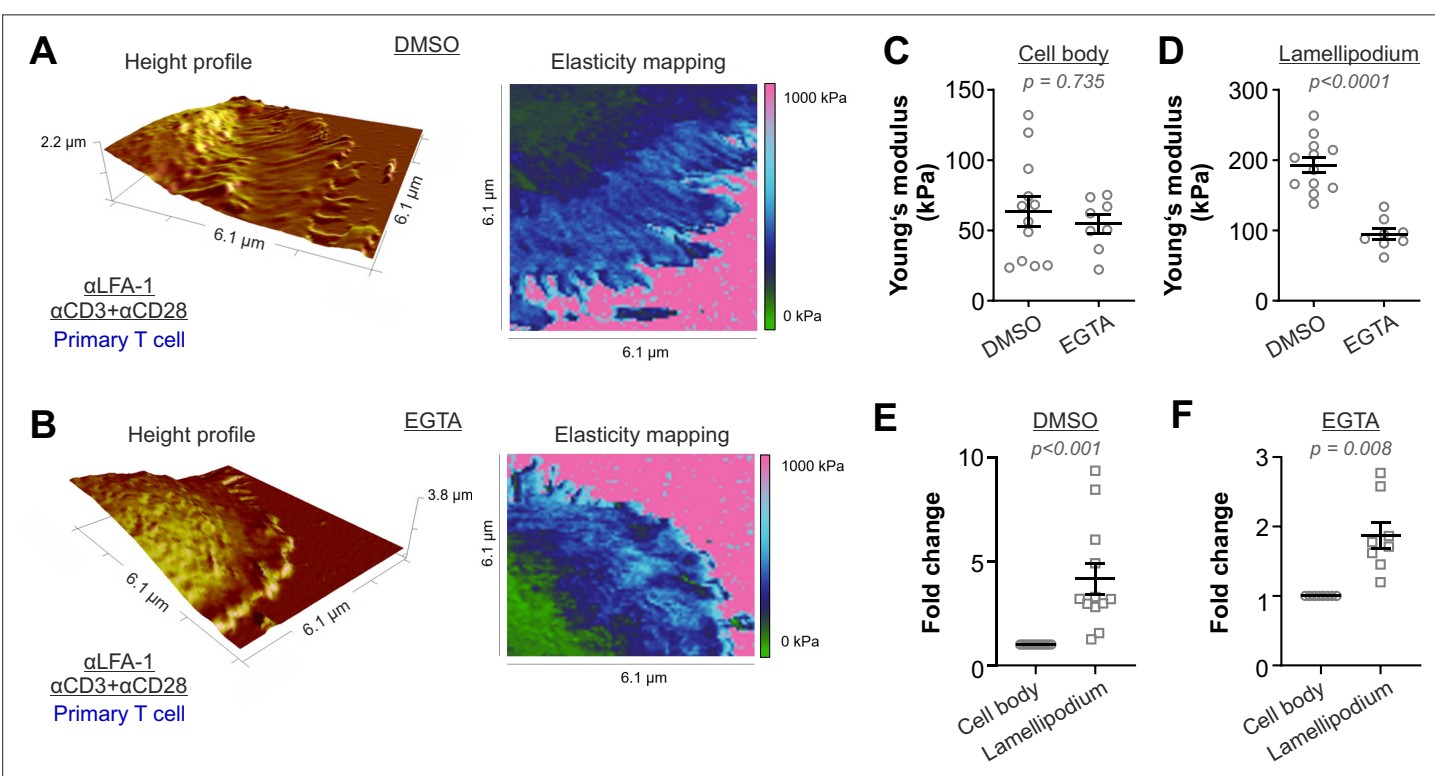

**Figure 3.** Activation-induced T cell stiffening is regulated by intracellular calcium. Primary human CD4+ T cells were treated with either EGTA-AM or DMSO at room temperature for 30 min. Height profiles and corresponding elasticity maps (**A, B**), Young's moduli (**C, D**), and the respective fold changes (**E, F**) are shown. The Mann-Whitney test (**C, D**) or the Wilcoxon matched-pairs signed rank test (**E, F**) was used for statistical significance. Results were presented as mean ± SEM, from 11 cells for each condition (LFA-1 vs full anybody-set) from four independent experiments (four donors). Source data please refer to *Figure 3—source data 1*. For height profiles and elasticity maps of each value and condition, refer to *Figure 3—figure supplements 1 and 2*.

The online version of this article includes the following source data and figure supplement(s) for figure 3:

**Source data 1.** Original values of stiffness shown in *Figure 3*.

**Figure supplement 1.** Height profiles and elasticity maps (Young`s modulus) of DMSO-treated primary T cells on full antibody (αLFA-1+αCD3+CD28)-functionalized glass.

**Figure supplement 2.** Height profiles and elasticity maps (Young`s modulus) of EGTA-treated primary T-cells on full antibody (αLFA-1+αCD3+CD28)-functionalized glass.

*2016*), we first targeted the actin-cytoskeleton with latrunculin-A (actin polymerization inhibitor) as well as the microtubule-network with nocodazole (a microtubule depolymerizing agent). Not surprisingly, with the disassembly of the cytoskeleton, T cells failed to attach to the functionalized surface firmly enough for AFM measurements. Next, we turned our focus to $Ca^{2+}$. We chelated intracellular $Ca^{2+}$ with EGTA-AM (dissolved in DMSO) and found that the lamellipodia formed by $Ca^{2+}$-chelated $CD4^+$ T cells exhibited a substantially lower stiffness compared to their vehicle (DMSO)-treated counterparts (*Figure 3A–D*). Notably, the stiffness of the cell body was not affected by $Ca^{2+}$ chelation (*Figure 3C*). In both DMSO and EGTA-treated conditions, the stiffness of lammellipodia was higher than that of the cell body (*Figure 3E and F*). These results indicate that $Ca^{2+}$ is involved in regulating stiffening of the lamellipodia but not of the cell body upon IS formation.

## Discussion

It has previously been reported that when extracellular $Ca^{2+}$ is chelated by EGTA, TCR activation can still induce a transient and moderate elevation of intracellular $Ca^{2+}$, which is sufficient to initiate some downstream events including actin polymerization and actin-dependent spreading (*Babich and Burkhardt, 2013*). However, such a transient and moderate elevation of intracellular $Ca^{2+}$ would substantially inhibit the release of cytotoxic granules as shown in very low extracellular $Ca^{2+}$ condition (3 μM) (*Zhou et al., 2018*). Therefore, our observation that chelation of intracellular $Ca^{2+}$ reduced the stiffness only at the lamellipodia but not the cell body suggests that sustained or higher levels of intracellular $Ca^{2+}$ might be required to stiffen lamellipodia, and the stiffened lamellipodia could be important for vesicle fusion at the IS. We postulate that local rearrangement of cytoskeleton might contribute to a large extent to this $Ca^{2+}$ dependent stiffening at lamellipodia induced by IS formation. On one hand, integrity of actin-cytoskeleton is essential in T cells to generate mechanical forces at the IS (*Basu et al., 2016*; *Fritzsche et al., 2017*). On the other hand, fast growth of microtubules in $CD4^+$ T cells at the contact site between anti-CD3 antibody-coated coverslip is observed, and the traction stresses at the IS generated by actomyosin contractility is increased after disassembly of the microtubule-network (*Hui and Upadhyaya, 2017*). In addition, cytoskeleton-regulatory proteins such as ROCK and cofilin might be involved in this $Ca^{2+}$ dependent local stiffening (*Butte et al., 2014*; *Thauland et al., 2017*).

In our work, the range of the determined Young's moduli is considerably higher than some of the previously reported ones, which were in a range of tens to hundreds of Pa (*Butcher et al., 2009*). However, it is important to note that the studies discussed in the aforementioned review used nanoindentation experiments to determine the Young's moduli. Compelling evidence in recent years shows that AFM experiments are influenced by a multitude of parameters, such as the subsurface material, the choice of the cantilever, the contact model applied, environmental conditions, and the measurement mode. In our work, the Peak Force QNM mode, also known as Peak Force Tapping mode, was applied, which has the advantage to dynamically modulate the *z* piezo below the cantilever resonance frequency, allowing detailed mapping of cells in a reduced amount of time. Unfortunately, the direct comparability of Young's moduli obtained with this method to elasticity data obtained with classical nanoindentation methods, which reported Young's moduli in the Pa range, seems to be mostly lost. However, the Young's moduli reported here for T cells are in good agreement to other elasticity data published for different human cell types with the Peak Force QNM mode. For instance, a work investigated the glyphosate induced stiffening of human keratinocytes (HaCaT) by applying the Peak Force QNM mode. Here, Young's moduli of approximately 50–300 kPa in HaCaT were determined (*Heu et al., 2012*). Another study utilized the Peak Force QNM mode to address the role of cholesterol assemblies on the mechanical behavior of mammalian breast cancer cells (MCF10), and observed Young's moduli of approximately 5–44 kPa (*Dumitru et al., 2020*). Calzado-Martin et al. studied the effect of actin organization on the stiffness of breast cancer cells lines by Peak Force QNM mode, which revealed Young's moduli of approximately 50–150 kPa (*Calzado-Martín et al., 2016*). Interestingly, the reduction of the Peak Force QNM modulation frequency from 250 Hz to 1 Hz resulted in a tremendous decrease in the absolute values of Young's moduli of more than 2 orders of magnitude, which further emphasizes the impact of varying measurement parameters during elasticity mapping (*Calzado-Martín et al., 2016*). A recent review by *Li et al., 2021* emphasizes the technical improvements and advantages of the Peak Force QNM mode and specifically recommends this AFM mode for immunological applications.

Previous approaches to study the stiffness of T cells utilized among others microplate and micro-manipulation techniques, and reported Young's moduli of around 100 Pa (*Bufi et al., 2015*) and 50 kPa (*Du et al., 2017*), respectively. For the microplate approach, the contact area between the flexible microplate and the T-cell is considerably large, close to the diameter of the whole cell. Earlier AFM approaches carried out to determine the stiffness of immune cells utilized a glass or silicon sphere (diameter around 1–5 μm) attached to the cantilever to measure cell stiffness, and reported Young's moduli in the range of a few hundred Pa (*Sadoun et al., 2021*) to several thousand Pa (*Blumenthal et al., 2020*). In contrast, we used cantilevers with a pyramidal, rounded tip (diameter: ~ 60 nm). Considering that microbead pillows are very soft but microbeads per se are stiff, cell stiffness measured from a larger scale could differ from its local microscale stiffness. Of note, the methods used to measure cell stiffness in a larger scale is not suitable to determine stiffness of lamellipodial regions.

When scanning the vicinities of the attached T cells, we noticed that some points on glass coverslips were particularly soft (around 100–400 kPa). Thus, we carefully compared the elasticity mapping and the height profiles, and found that most soft points from the elasticity mapping overlap with small 'bumps' in the height profiles, which seem to be connected to the lamellipodia with thin fibers (e.g. *Figure 2A and B*). These small bumps might be cell debris left on the surface after retraction of lamellipodia as shown in lamellipodial dynamics in *Figure 1A* (compare 40 min to 20 min).

## Materials and methods

### Key resources table

| Reagent type (species) or resource | Designation | Source or reference | Identifiers | Additional information |
|---|---|---|---|---|
| Cell line (*Homo sapiens*) | Jurkat E6.1 cell line | ATCC | ATCC Cat# TIB-152, RRID:CVCL_0367 | |
| Biological sample (*Homo sapiens*) | Primary human CD4+ T cells | Human peripheral blood mononuclear cells (PBMCs) were obtained from healthy donors provided by Institute of Clinical Hemostaseology and Transfusion Medicine. Faculty of Medicine. University of Saarland. PMID:24599783 | | Negatively isolated from PBMCs using CD4+ T Cell Isolation Kit human (Miltenyl). |
| Commercial assay or kit | CD4+ T Cell Isolation Kit human | Miltenyi | Cat# 130-096-533 | |
| Commercial assay or kit | Sylgard 184 Silicone Elastomer Kit | Dow Europe GmbH | Material Number 1317318 | |
| Peptide, recombinant protein | Polyornithine | Sigma-Aldrich (Merck) | MDL number MFCD00286305 | |
| Chemical compound, drug | EGTA/AM | Calbiochem (Merck) | Cat# 324,628 | |
| Antibody | anti-LFA-1 (ITGAL) antibody (Mouse monoclonal) | Antibodies-online | Cat# ABIN135680, RRID:AB_10773722 | Diluted to 9 μg/ml in 20 μl PBS |
| Antibody | mouse anti-human CD28 antibody (Mouse monoclonal) | BD Pharmingen | Cat# 555725, RRID:AB_396068 | Diluted to 90 μg/ml in 20 μl PBS |
| Antibody | mouse anti-human CD3 antibody (Mouse monoclonal) | Diaclone | Cat# 854.010.000, RRID:AB_1155287 | Diluted to 30 μg/ml in 20 μl PBS |
| Software, algorithm | GraphPad Prism | GraphPad | RRID:SCR_002798 | |
| Software, algorithm | Research NanoScope 9.1 | Bruker Corp. | R3.119071 | |
| Software, algorithm | NanoScope Analysis 1.80 | Bruker Corp. | R2.132257 | |

## Antibodies and reagents

All chemicals not specifically mentioned are from Sigma-Aldrich (highest grade). The following antibodies or reagents were used: anti-LFA-1 (ITGAL) antibody (Antibodies-online), mouse anti-human CD28 antibody (BD Pharmingen), and mouse anti-human CD3 antibody (Diaclone).

## Cell lines

The Jurkat T-cell line (E6.1) was purchased from ATCC. We confirm that no mycoplasma contamination is detected by regular examinations.

## Cell culture

Human peripheral blood mononuclear cells (PBMCs) were obtained from healthy donors as described before (*Kummerow et al., 2014*). Primary human CD4[+] T cells were negatively isolated from the PBMCs using CD4[+] T Cell Isolation Kit human (Miltenyl) and cultured in AIM V medium (ThermoFisher Scientific) with 10 % FCS (ThermoFisher Scientific). Jurkat T-cells were cultured in RPMI-1640 medium (ThermoFisher Scientific) with 10 % FCS. All cells were cultured at 37 °C with 5 % $CO_2$.

## Preparation of antibody-functionalized surface

The glass coverslips or PDMS (2.5 MPa) were first coated with Polyornithine at room temperature for 1 hour. The concentrations of anti-LFA-1/anti-CD3/anti-CD28 antibodies are: 9 µg/ml, 30 µg/ml, and 90 µg/ml, respectively. The antibodies as indicated were coated either at 37 °C for 30 min or at 4 °C overnight.

## AFM-based elasticity mapping in combination with light microscopy

Microscopic observation during elasticity mapping was carried out on a DMI 4000 B inverted microscope (Leica) with a 200-fold magnification. Cells were first settled on coverslips at 37 °C with 5 % $CO_2$ for 15 min (*Berquand et al., 2010*). Prior to each experiment, the AFM cantilever (MLCT cantilever B, Bruker) was calibrated using the thermal tune method (*Li et al., 2020*). The spring constant given by manufacturer is 0.02 N/m (min 0.005, max 0.04 N/m), and the calibrated spring constant is 0.06–0.1 N/m, which varies slightly in different cantilevers. Elasticity mapping using a Bioscope Catalyst (Bruker) in Peak Force Quantitative Nanoscale Mechanical Characterization mode (Peak Force QNM) (*Berquand et al., 2010*) was conducted in cell culture media and carried out with the following parameters: line scan rate: 0.25 Hz, feedback gain: 0.5, peak force amplitude: 100 nm, peak force threshold: 700 pN and a resolution of ~60 nm. Young's moduli were obtained by a Derjaguin-Muller-Toporov (DMT) fit (*Derjaguin et al., 1975*) of the retract part of each single force/distance curve. Elasticity maps (square-shaped with side length of 5–10 µm) spanning approximately a quarter of the cell, including lamellipodium and cell body, were recorded. Elastic moduli of the T-cells were determined as square shaped surface segments located on the cell bodies and lamellipodia. For lamellipodia, three individual square-shaped surface segments of 500 × 500 nm were analyzed per cell. If very slender filopodia structures with a lateral width of less than 500 nm were seen, the analyzed segment size was reduced to 250 × 250 nm. To determine the elastic moduli of the cell bodies, one 1.5 × 1.5 µm surface segment of the peripheral region was investigated per cell. Approximately 29,600 individual elasticity values were analyzed on a total of 58 primary T lymphocytes. Representative force curves and elasticity maps for each condition are provided in figure supplements. The stiffness of uncoated glass coverslips was determined in air using the cantilevers (ScanAsyst Air, Bruker) with a spring constant of 0.8 N/m and the following parameters: line scan rate: 0.5 Hz, feedback gain: 1.5, peak force amplitude: 100 nm, peak force threshold: 6 nN and a resolution of ~2 nm.

## Elastomer production

Polydimethylsiloxane (PDMS) elastomers with an AFM-validated stiffness of 2.5 MPa were produced by crosslinking the base component (A) methylhydrosiloxane-dimethylsolioxane with the crosslinking reagent vinyl-terminated polymethylsiloxane (Component B; both Sylgard 184 Silicone Elastomer Kit, Dow Europe GmbH). An electrostatic deionizer (Eltex-Elektrostatik GmbH) was used to remove static charges from the materials prior to use with PDMS. Component A and B were mixed at a 10:1 ratio in polypropylene tubes, mixed vigorously, and de-gassed by centrifugation. The PDMS mixture was

poured into the lid of 50 mm cell culture dishes, left for 1 hr to settle before being placed in a 60 °C incubator to cure for 16 hours. PDMS elastomers were then functionalized as described above.

## Statistical analysis

Data are presented as mean ± SEM. GraphPad Prism 6 Software (San Diego, CA, USA) was used for statistical analysis. The differences between two groups were analyzed by either the Mann-Whitney test (unpaired test, not assuming Gaussian distribution) or the Wilcoxon matched-pairs signed rank test (paired test, not assuming Gaussian distribution) as indicated in the figure legends. p-Values < 0.05 were considered significantly different.

## Acknowledgements

We thank the Institute for Clinical Hemostaseology and Transfusion Medicine for providing donor blood; Carmen Hässig, Cora Hoxha, Gertrud Schäfer, Sandra Janku, and Mengnan Li for excellent technical help. This project was funded by the Deutsche Forschungsgemeinschaft (SFB 1027 projects A2 to BQ, B2 to MB, A12 to SI, and SPP1782 ID79/2-2 to SI), INM Fellow (to BQ).

## Additional information

### Funding

| Funder | Grant reference number | Author |
| --- | --- | --- |
| Deutsche Forschungsgemeinschaft | SFB1027 A2 | Bin Qu |
| Deutsche Forschungsgemeinschaft | SFB1027 B2 | Markus Bischoff |
| Deutsche Forschungsgemeinschaft | SPP1782 ID79/2-2 | Sandra Iden |
| Leibniz-Gemeinschaft | INM Fellowship | Bin Qu |
| Deutsche Forschungsgemeinschaft | SFB1027 A12 | Sandra Iden |

The funders had no role in study design, data collection and interpretation, or the decision to submit the work for publication.

### Author contributions

Philipp Jung, Formal analysis, Investigation, Methodology, Validation, Visualization, Writing – review and editing; Xiangda Zhou, Investigation, Methodology, Writing – review and editing; Sandra Iden, Methodology, Resources, Writing – review and editing; Markus Bischoff, Formal analysis, Funding acquisition, Methodology, Resources, Writing – review and editing; Bin Qu, Conceptualization, Funding acquisition, Investigation, Resources, Supervision, Writing – original draft, Writing – review and editing

### Author ORCIDs

Philipp Jung ⓘ http://orcid.org/0000-0001-6182-6592
Sandra Iden ⓘ http://orcid.org/0000-0003-2333-9827
Bin Qu ⓘ http://orcid.org/0000-0002-9382-3203

### Decision letter and Author response

Decision letter https://doi.org/10.7554/eLife.66643.sa1
Author response https://doi.org/10.7554/eLife.66643.sa2

## Additional files

### Supplementary files
• Transparent reporting form

### Data availability
All data generated or analysed during this study are included in the manuscript, figure supplements or source data files. All files are uploaded.

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
