## [Decision Letter]

**Acceptance summary:**

The mechanics of lymphocytes, such as T cells, remains relatively less explored as compared to tissue-cells that are now relatively well known. This excellent work will undoubtedly be of interest to immunologists and scientists interested in cell mechanics.

**Decision letter after peer review:**

Thank you for submitting your article "T cell stiffness is enhanced upon formation of immunological synapse" for consideration by *eLife*. Your article has been reviewed by 3 peer reviewers, and the evaluation has been overseen by a Reviewing Editor and Anna Akhmanova as the Senior Editor. The following individual involved in review of your submission has agreed to reveal their identity: Alphee Michelot (Reviewer #1).

Essential revisions:

1. The elasticity maps scale in MPa range. The values for the Young's moduli are surprisingly high in the range of hundreds of kPa, suggesting that the underlying glass surface might have influenced the absolute values.

The authors should provide compelling arguments to prove that the observed stiffening of the lamellipodium and cell body of T cells is free of influence from the glass surface.

Ideally, the authors should confirm their claims by repeating the measurements on antibody functionalised hydrogels (in the MPa range to limit the effects of mechanosensation) for resting and activating Jurkat or primary CD4^+^ T cells (or any other means in the spirit of the hydrogel to exclude the influence of the substrate).

2. The manuscript often lacks details. Figure Legends are too succinct. A clear Legend for each panel of each Figure is necessary.

Also, could the authors please provide additional information in the statistics statement in the method section? Information about the number of cells and independent experiments performed are hard to find. The precise number of measurements, experiments and biological replicates need to be reported for each data set. Representative force curves and histograms of all values as supplementary data should be included. The choice of the two kinds of statistical tests should be briefly elaborated.

3. The authors should make full use of the spatial high resolution of the stiffness maps in space (instead of quantifying only small 500µm x 500 µm regions). It might be very helpful to understand how the mechanical environment influences locally the T cell stiffness upon activation. Zooms into the stiffness maps and correlations to the cell protrusions may uncover further detail, and thus strengthen the breadth of the study.

*Reviewer #1:*

In this paper, the authors investigate T cell stiffness changes upon CD3/CD28 activation. Jurkat T cells or Primary T cells are plated on surfaces coated with the adhesion molecule LFA-1, in the presence or in the absence of CD3/CD28. Cell shape and stiffness are probed with AFM.

Results indicate a clear stiffening of these cells upon CD3/CD28 activation. Interestingly, calcium chelation reduces stiffening of lamellipodia but not of the cell body, suggesting different regulatory mechanisms.

The observations seem interesting and solid, though additional experimental support would make the paper stronger.

*Reviewer #2:*

The mechanics of lymphocytes, such as T cells, remains relatively less explored as compared to tissue-cells that are now relatively well know. The authors use AFM to map the elasticity of whole T cells at sub-micron scales, using both Jurkat and primary cells. They relate the cell-stiffness to the activation state of the cell. They find a calcium dependent stiffening response upon spreading of the cell to form an immune synapse.

These results will be of great interest to both immunologists and the cell mechanics community.

*Reviewer #3:*

Jung et al. employ AFM measurements to quantify cellular architectures of Jurkat and primary CD4^+^ T cells during antibody-mediated activation. The authors report changes in the stiffness of the cells, which are abolished in the absence of intra-cellular calcium release.

While the study on the influence of the mechanical environment on T-cell activation is important and timely, in my view, the authors do not provide convincing enough evidence for their claims, and thus the findings appear problematic without further controls. The present study reports high values for the stiffness of resting and activated Jurkat and primary T cells compared to other cell types (Ahmad et al., Micro and Nano Systems Letters 2014, Mandriota et al., Nature Methods 2019), highlighting the need for appropriate controls.

[Editors' note: further revisions were suggested prior to acceptance, as described below.]

Thank you for resubmitting your work entitled "T cell stiffness is enhanced upon formation of immunological synapse" for further consideration by *eLife*. Your revised article has been evaluated by Anna Akhmanova (Senior Editor) and a Reviewing Editor.

We discussed with the Reviewers and agree that your manuscript has been improved. However, we also think that a remaining issue, related to our previous point 1/, needs to be further addressed. We are convinced that comparisons made in your manuscript are fair, but we are not convinced that absolute values are fully trustworthy. Your values are rather high compared to what is published in the literature, and still do not allow to formally exclude an effect from the surface. If possible, we would like to lift this ambiguity in order to convince people in the field (which would also benefit you).

We thought that experiments on softer surfaces would bring an answer to this problem, but it seems that having a support of 2.5 MPa is still equivalent to having glass (GPa). So the new experiments do not really address the concern about "feeling" the substrate underneath. We apologize if our previous decision letter wasn't clear enough, but we thought that you would reduce substrate rigidity down to values that are appropriate to draw unambiguous conclusions. Could you please address this problem in a revised manuscript?

[Editors' note: further revisions were suggested prior to acceptance, as described below.]

Thank you for resubmitting your work entitled "T cell stiffness is enhanced upon formation of immunological synapse" for further consideration by *eLife*. Your revised article has been evaluated by Anna Akhmanova (Senior Editor) and a Reviewing Editor.

The manuscript has been improved but there are some remaining issues that need to be addressed, as outlined below. These modifications should not need the acquisition of new data, but mainly changes in the presentation:

1. Please include your AFM data on 400 kPa PDMS substrates in Figure 2 and as a Supplementary Figure, like you did for the other surfaces. You also need to review your Figure Legends which mention sometimes only "PDMS", without making clear to readers whether it corresponds to 400 kPa or 2.5 MPa PDMS. In Figure 2M, please write the cell type as you did in the other panels and specify the PDMS stiffness.

2. Your statements line 233/234 that "Apart from these particularly soft spots, the general Young's moduli measured on the glass coverslips are in the range of several to 10Mpa" and line 241/242 "Noticeably, on the functionalized PDMS surface, Young's moduli ranged from a few hundred kPa to 2.16 Mpa, similar to our observations made on the functionalized glass coverslips" seem contradictory. It is obvious from many images (e.g. Figure 2G) that Young moduli measured on functionalized glass coverslips is sometimes well below 1 MPa. Does your statement line 233/234 correspond to experiments with cells on non-functionalized surfaces that are not present in this manuscript?

It is clear that surface coating can very well influence the apparent rigidity of the surface and it is necessary to quantify this point unambiguously. I say that because Figure 2, for example, gives a visual impression that glass surfaces with αLFA-1 are generally softer than surface with αLFA-1+αCD3+αCD28 (which is probably wrong, but impossible to determine from the images of Supplementary Figure 2 which are plotted with different color scales). As substrate rigidity has been a matter of debate in this review, could you please provide dotplots of measured substrate rigidities (from your exisiting data) for the different conditions (glass, PDMS 2.5 MPa and 400 kPa, non-functionalized or functionalized with αLFA or αLFA-1+αCD3+αCD28)?

---

## [Author Response]

Essential revisions:1. The elasticity maps scale in MPa range. The values for the Young's moduli are surprisingly high in the range of hundreds of kPa, suggesting that the underlying glass surface might have influenced the absolute values.The authors should provide compelling arguments to prove that the observed stiffening of the lamellipodium and cell body of T cells is free of influence from the glass surface.Ideally, the authors should confirm their claims by repeating the measurements on antibody functionalised hydrogels (in the MPa range to limit the effects of mechanosensation) for resting and activating Jurkat or primary CD4^+^ T cells (or any other means in the spirit of the hydrogel to exclude the influence of the substrate).

We thank the reviewer for this constructive suggestion. We agree that the stiffness of underlying materials might influence the results of AFM-based elasticity mapping. To address this concern, we followed the reviewer’s advice and used PDMS substrates with a stiffness of 2.5 MPa (rev. Figure 2M). The stiffness of primary T cells was characterized on a full antibody set (αLFA-1+αCD3+αCD28)-functionalized PDMS substrate using the same settings as for glass substrates. As shown in figure 2, no significant difference was observed between glass and PDMS for the cell body or at the lamellipodium (rev. Figure 2N,O). It indicates that the Young's moduli in the range of hundreds of kPa after activation is not due to the influence from glass substrates. This conclusion is further supported by our observation shown in rev. Figure 2 (A-D) in the manuscript, that at lamellipodia or cell body, the Young's moduli are in the range of tens of kPa in the control condition (αLFA-1).

This can be explained by the ratio between the thickness of the biomaterial layer and the indentation depth during elasticity mapping. We measured the mean thickness of the lamellipodium and the cell body which is 144 ± 69.7 nm and up to several µm, respectively. During elasticity mapping a peak force threshold of 700 pN was applied which caused a comparably small indentation of 17 nm (± 4.7) into the material of the lamellipodium and 21 nm (± 2.2) into the cell body. This is less than 12 % (lamellipodium) and 0.8 % (cell body) of the biomaterial layer. We can therefore largely exclude an impact of the underlying surface on the measured T cell stiffness. This information is part of the Results and Discussion section of the revised version of the manuscript (revised Figure 2M-O, text on Page 6).

2. The manuscript often lacks details. Figure Legends are too succinct. A clear Legend for each panel of each Figure is necessary.Also, could the authors please provide additional information in the statistics statement in the method section? Information about the number of cells and independent experiments performed are hard to find. The precise number of measurements, experiments and biological replicates need to be reported for each data set. Representative force curves and histograms of all values as supplementary data should be included. The choice of the two kinds of statistical tests should be briefly elaborated.

We thank the reviewer for pointing this out. As suggested, the information about the number of cells/measurements and biological replicates have been added to the figure legends. Concerning force curves, the single Force Distance Curves were not saved in PeakForce QNM for the datasets in the original submission. However, since the reviewer pointed out, for the new experiments done for αLFA-1+αCD3+αCD28 functionalized-PDMS surface, we saved the curves and representative force curves are provided in Figure2—figure supplement 6. Elasticity maps for each cell are provided in Figure 2—figure supplement 2-5 and Figure 3—figure supplement 1-2. The choice of the statistical tests is explained in the Materials and methods section as follows:

“The differences between two groups were analyzed by either the Mann-Whitney test (*unpaired test, not assuming Gaussian distribution*) or the Wilcoxon matched-pairs signed rank test (*paired test, not assuming Gaussian distribution*) as indicated in the figure legends.”

3. The authors should make full use of the spatial high resolution of the stiffness maps in space (instead of quantifying only small 500µm x 500 µm regions). It might be very helpful to understand how the mechanical environment influences locally the T cell stiffness upon activation. Zooms into the stiffness maps and correlations to the cell protrusions may uncover further detail, and thus strengthen the breadth of the study.

We followed the reviewer’s advice and analyzed a bigger selection of individual spots on the lamellipodia, especially at tips/edges, close to the cell body, and in between (see Author response image 1, Young’s moduli are given individually). Here, we observed very similar Young’s moduli for these regions from the same cell, with (αLFA-1+αCD3+αCD28, the right column) or without (αLFA-1, the right column) the IS. This finding indicates that the elasticity within the lamellipodia is rather a universal property, which is not affected by the topology or the position of the micro-structures with an area of a few hundred nm^2^. Notably, cytoskeleton serves as scaffolds at the protrusions such as actin filaments or microtubules. The diameters of these filamentous structures (~ 5-25 nm) are, however, far below the resolution (~ 60 nm) of our method. Thus, we cannot exclude the possibility that along these cytoskeletal filaments the elasticity differs from the neighboring structures. This information is part of the Results and Discussion section (revised Figure 2—figure supplement 1, text on Page 6).

**Author response image 1. sa2fig1:** Elasticity maps and height profiles of representative cells of each substrate condition.

[Editors' note: further revisions were suggested prior to acceptance, as described below.]

We discussed with the Reviewers and agree that your manuscript has been improved. However, we also think that a remaining issue, related to our previous point 1/, needs to be further addressed. We are convinced that comparisons made in your manuscript are fair, but we are not convinced that absolute values are fully trustworthy. Your values are rather high compared to what is published in the literature, and still do not allow to formally exclude an effect from the surface. If possible, we would like to lift this ambiguity in order to convince people in the field (which would also benefit you).

We fully agree with the reviewers/editor about this point. To further investigate the possibility of Young’s moduli being influenced by the stiffness of the underlying glass coverslip, we conducted as suggested additional experiments with primary T cells on αLFA-1+αCD3+αCD28 functionalized PDMS substrates with an elasticity of ~400 kPa. Even on this considerably softer substrate, no significant changes were identified in the Young`s moduli of the lamellipodia of T cells, compared to the ones measured on glass or on stiffer PDMS substrates (~2.5 MPa). The new results are integrated in the manuscript as revised Figure 2—figure supplement 2.

Of course, we understand the reviewer`s concern very well that the range of the Young`s moduli presented in our manuscript is much higher than the reported ones (in a range of tens to hundreds of Pa). We would like to take the opportunity to further discuss this issue here:

One of the most influential review written by Butcher and colleagues in 2009 (cited > 1500), summarized elastic values of different tumor cell types and tissues, most of which are below 5 kPa (Butcher et al., 2009).

We thought that experiments on softer surfaces would bring an answer to this problem, but it seems that having a support of 2.5 MPa is still equivalent to having glass (GPa). So the new experiments do not really address the concern about "feeling" the substrate underneath. We apologize if our previous decision letter wasn't clear enough, but we thought that you would reduce substrate rigidity down to values that are appropriate to draw unambiguous conclusions. Could you please address this problem in a revised manuscript?

However, it is important to note that the publications cited in this review had used nanoindenation experiments to collect data, since this was the only way to address elasticity on a subcellular level by AFM in 2009 and before. For nanoindentation experiments, the work from Butcher *et al.* can still serve as an excellent orienting scale to classify elastic moduli of living cells.

Remarkably, compelling evidence in recent years show that AFM experiments are influenced by a multitude of parameters, such as the subsurface material, the choice of the cantilever, the contact model applied, environmental conditions, and the measurement mode. Here, the Peak Force Tapping mode, initially introduced in 2010 and applied in our work, has the advantage to dynamically modulate the z piezo below the cantilever resonance frequency, which allows detailed mapping of cells in a reduced amount of time. However, the comparability to classical nanoindentation data seems to be mostly lost, as shown in the other studies. For instance, a work investigated the Glyphosate induced stiffening of human keratinocytes (HaCaT) by applying the Peak Force Tapping mode. Here, Young`s moduli of approximately 50 to 300 kPa in HaCaT were determined (Heu, Berquand et al., 2012). Another study utilized the Peak Force Tapping mode to address the role of cholesterol assemblies on the mechanical behavior of mammalian breast cancer cells (MCF10), and observed Young`s moduli of approximately 5 to 44 kPa (Dumitru, Mohammed et al., 2020). Calzado-Martin *et al.* studied the effect of actin organization on the stiffness of breast cancer cells lines by Peak Force Tapping, which revealed Young`s moduli of approx. 50 kPa to 150 kPa (Calzado-Martin, Encinar et al., 2016). Interestingly, the reduction of the Peak Force Tapping modulation frequency from 250 Hz to 1Hz resulted in a tremendous decrease in the absolute values of Young`s moduli of more than 2 orders of magnitude, which further emphasizes the impact of varying measurement parameters during elasticity mapping (Calzado-Martin et al., 2016). With all limitations in comparability, the Young’s moduli presented in our study are in a comparable range as described in the studies mentioned above, where the Peak Force Tapping mode was utilized. Therefore, we believe that a comparability between AFM results is only given, if very similar methodologies were applied, and even then the variation of single measurement parameters may have a tremendous impact on the Young`s moduli obtained. Of note, a recent review published in early 2021 by Li *et.al.* emphasizes the technical improvements and advantages of the Peak Force Tapping mode and specifically recommends this AFM mode for immunological applications (Li, Liu et al., 2021). Previous approaches to study the stiffness of T cells had utilized microplate and micromanipulation techniques, with Young’s moduli of around 100 Pa (Bufi, Saitakis et al., 2015) and 50 kPa (Du, Kalia et al., 2017), respectively. For the microplate approach, the contact area between the flexible microplate and the T-cell is considerably large, close to the diameter of the whole cell. For AFM approaches, in some studies a glass or silicon sphere (diameter around 1-5 µm) was attached to the cantilever to measure immune cell stiffness, with Young’s moduli in the range of a few hundred Pa (Sadoun, BiarnesPelicot et al., 2021) to several thousand Pa (Blumenthal, Chandra et al., 2020). In contrast, we used cantilevers with a pyramidal, rounded tip (diameter: 60 nm). Considering that microbead pillows are very soft but microbeads per se are stiff, cell stiffness measured from a larger scale could differ from its local micro-scale stiffness.

Taken together, we are convinced that the values reported here are reliably determined by the state-ofthe-art AFM approach with a high resolution (~60 nm). This high resolution is especially required for determination of lamellipodial stiffness, which cannot be measured by AFM with bigger sphere-shaped tips (1-5 µm) or the microplate technique. Therefore, our findings presented here will be of great interest for the scientific community.

Given the importance of this point, we have included it in Discussion section in Page 9-10.

[Editors' note: further revisions were suggested prior to acceptance, as described below.]

The manuscript has been improved but there are some remaining issues that need to be addressed, as outlined below. These modifications should not need the acquisition of new data, but mainly changes in the presentation:1. Please include your AFM data on 400 kPa PDMS substrates in Figure 2 and as a Supplementary Figure, like you did for the other surfaces. You also need to review your Figure Legends which mention sometimes only "PDMS", without making clear to readers whether it corresponds to 400 kPa or 2.5 MPa PDMS. In Figure 2M, please write the cell type as you did in the other panels and specify the PDMS stiffness.

As suggested, one representative cell measured on 400 kPa PDMS substrates has been added as revised Figure 2L and the original data are summarized in Figure 2—figure supplement 6. The cell type used was specified in the figures.

2. Your statements line 233/234 that "Apart from these particularly soft spots, the general Young's moduli measured on the glass coverslips are in the range of several to 10Mpa" and line 241/242 "Noticeably, on the functionalized PDMS surface, Young's moduli ranged from a few hundred kPa to 2.16 Mpa, similar to our observations made on the functionalized glass coverslips" seem contradictory. It is obvious from many images (e.g. Figure 2G) that Young moduli measured on functionalized glass coverslips is sometimes well below 1 MPa. Does your statement line 233/234 correspond to experiments with cells on non-functionalized surfaces that are not present in this manuscript?

What we meant to state in line 233/234 is "Apart from these particularly soft spots, the general Young's moduli measured on the glass coverslips are in the range of several hundred kPa to 10 MPa". We overlooked this mistake. We sincerely apologize for this carelessness.

Concerning the old Figure 2G, we agree with the editor/reviewer that the Young moduli measured on functionalized glass coverslips are below 1 MPa. We examined the original data (summarized in revised Figure 2—figure supplement 2) and found out that in the example we chose for the old Figure 2G, the Young’s moduli of the functionalized substrates were not representative from all the cells measured (revised Figure 2—figure supplement 3). Therefore, we have now chosen a more representative cell for this condition in the revised figure (revised Figure 2A).

It is clear that surface coating can very well influence the apparent rigidity of the surface and it is necessary to quantify this point unambiguously. I say that because Figure 2, for example, gives a visual impression that glass surfaces with αLFA-1 are generally softer than surface with αLFA-1+αCD3+αCD28 (which is probably wrong, but impossible to determine from the images of Supplementary Figure 2 which are plotted with different color scales). As substrate rigidity has been a matter of debate in this review, could you please provide dotplots of measured substrate rigidities (from your exisiting data) for the different conditions (glass, PDMS 2.5 MPa and 400 kPa, non-functionalized or functionalized with αLFA or αLFA-1+αCD3+αCD28)?

We thank the editor/reviewer for bringing up this important point. To address it, we firstly examined the stiffness of uncoated glass coverslips. Since the expected Young’s moduli are in the range of GPa, we used a stiffer cantilever with a spring constant of 0.8 N/m and experimental settings suitable to characterize hard substrates (Further details please see Methods). As shown in Figure 2—figure supplement 2(A), Young’s moduli of uncoated glass coverslips were around 1 GPa, verifying that with these experimental conditions our system is able to detect the stiffness in the GPa range. Next, we determined the Young’s moduli of uncoated substrates with a soft cantilever and the experimental settings applied to characterize the stiffness of T cells. The average Young’s moduli of uncoated glass coverslips were around 18 MPa with the highest values around 20 MPa (B), which is clearly below the stiffness values determined with the stiffer cantilever and suggestively the upper limit of the Young’s moduli that can be determined by these experimental conditions optimized for determination of T cell stiffness. Nevertheless, the Young’s moduli of uncoated PDMS substrates (2.5 MPa and 400 kPa) were in the expected range, indicating that up to 2.5 MPa the stiffness can be precisely determined by the corresponding experimental conditions. Then as suggested, we analyzed the substrate rigidity for each value of our existing datasets. Concerning the datasets for primary T cells, the functionalization of stiffer substrates (i.e. glass and 2.5 MPa PDMS) with αLFA-1 or αLFA-1+αCD3+αCD28 created surfaces with drastically reduced Young’s moduli of around 600-700 kPa (B), while the stiffness of the αLFA-1+αCD3+αCD28 functionalized softer PDMS (400 kPa) was comparable to the uncoated PDMS (400 kPa) (B). This indicates that antibody-functionalization substantially decreases the apparent stiffness of the cantilever-accessible upmost surface layer of stiff substrates. These new data are included in the manuscript as revised Figure 2—figure supplement 2 and the corresponding text in Page 8-9.

For the dynamics of lamellipodia, we performed additional experiments with primary T cells. Dynamic changes of lamellipodia were also observed. We have included these new datasets as revised Figure 1B. Furthermore, the elasticity maps shown in the Supplemental Figures were revised to share the same data scale.